# Exploring the Therapeutic Potential of *trans*-Chalcone: Modulation of MicroRNAs Linked to Breast Cancer Progression in MCF-7 Cells

**DOI:** 10.3390/ijms241310785

**Published:** 2023-06-28

**Authors:** Tatiana Takahasi Komoto, Felipe Garcia Nishimura, Adriane Feijó Evangelista, Ana Julia Aguiar de Freitas, Gabriel da Silva, Wilson Araujo Silva, Kamila Peronni, Marcia Maria Chiquitelli Marques, Mozart Marins, Ana Lucia Fachin

**Affiliations:** 1Biotechnology Unit, University of Ribeirão Preto, SP, Av. Costábile Romano, 2201, Ribeirão Preto 14096-900, Brazil; tattytk@hotmail.com (T.T.K.); felipegnishi@hotmail.com (F.G.N.);; 2Molecular Oncology Research Center, Barretos Cancer Hospital, Teaching and Research Institute, Barretos 14784-400, Brazil; 3Sergio Arouca National School of Public Health, Oswaldo Cruz Foundation, Manguinhos, Rio de Janeiro 21040-361, Brazil; 4Center for Medical Genomics at the Clinics Hospital of Ribeirão Preto Medical School, University of São Paulo, Ribeirão Preto 14049-900, Brazil

**Keywords:** flavonoid, breast cancer, antitumor, microRNA, *trans*-chalcone

## Abstract

Breast cancer is responsible for 25% of all cancers that affect women. Due to its high heterogeneity pattern in clinical diagnosis and its molecular profile differences, researchers have been seeking new targets and therapies, with more specificity and fewer side effects. Thus, one compound that has garnered our attention is *trans*-chalcone, which is naturally occurring in various plants and possesses promising biological properties, including antitumor effects. MiRNA is an extensive class of non-coding small, endogenous, and single-stranded RNAs, and it is involved in post-translational gene regulation. Therefore, the objective of this study was to investigate the effects of TChal on miRNAs expression and its relationship with anticancer activity against MCF-7. Initially, the *trans*-chalcone IC50 value was established by MTT assay for MCF-7and HaCat (non-cancer cell), in which we found out that it was 53.73 and 44.18 μM, respectively. Subsequently, we treated MCF-7 cells with *trans*-chalcone at its IC50 concentration and performed Mi-seq analysis, which unveiled 23 differentially expressed miRNAs. From this set, we selected five miRNAs (miR-25-5p, miR-27a-3p, miR-891a, miR-449a, and miR-4485) for further validation using qRT-PCR, guided by in silico analysis and their known association with tumorigenesis. In conclusion, our research provides valuable insights into the potential use of TChal to reveal MicroRNAs molecular targets that can be applied in breast cancer therapy.

## 1. Introduction

Breast cancer is the most common type of cancer and accounts for 25% of all cancers affecting women. It is a heterogeneous disease at a molecular level [1], allowing it to be classified into five molecular subtypes: Luminal A, Luminal B HER-2 negative, Luminal B HER-2 positive, HER2 and basal-like tumors (basal-like or triple-negative) [2,3]. The complex nature of breast cancer, characterized by variations in clinical diagnosis and molecular profiles [4] has prompted researchers to explore novel targets and therapies that offer greater specificity and reduced side effects.

Flavonoid group has been outstanding in the research community due to its biological activities, for example, antioxidant effects, anti-aging properties, anti-inflammatory activity, immunomodulatory activity, cardioprotective effects, antiviral, antimicrobial, antifungal activity, and anti-cancer properties [5,6,7,8]. Among this group, *trans*-chalcone (TChal) has caught our attention, firstly because it is widely distributed in plants and secondly due to its biological properties. Our group has been dedicated to unraveling the mechanisms underlying the action of TChal. Our investigations have demonstrated the efficacy of TChal in combating breast cancer and osteosarcoma by targeting crucial genes involved in cancer development, such as *p53*, *Bcl2*, *AURKA*, and *MDR1* [9,10,11]. Furthermore, we have observed that TChal possesses the ability to suppress tumor growth mediated by Heme-Oxygenase 1 [12]. These findings highlight the potential of TChal as a promising therapeutic agent in the fight against cancer.

MiRNA is an extensive class of non-coding small RNAs that are endogenous, single-stranded, and 19–25 nucleotides in length. MiRNAs play a pivotal role in post-translational gene regulation, orchestrating a wide range of essential biological functions including proliferation, apoptosis, tumorigenesis, cellular differentiation, metastasis, and chemoresistance [13]. Because miRNAs are related to a variety of biological processes, they could be used as strategies in disease treatment and diagnosis as biomarkers or new targets for therapies [14,15].

Therefore, the present study aimed to investigate the TChal effects in miRNAs expression and its relationship with anticancer activity against breast cancer cell lines.

## 2. Results

### 2.1. trans-Chalcone Cytotoxicity

The cytotoxicity of TChal was evaluated against MCF-7, MDA-MB-231, and HaCaT (keratinocytes non-cancerous cell line) after 24 h exposure (Table 1). Using the established IC50 from MCF-7, the total RNA was extracted from MCF-7, treated and non-treated (0.1% DMSO) in three independent experiments. The bioanalyzer was then used to guarantee the RNA integrity (RIN > 7 for all samples - MCF-7 treated and non-treated with Tchal). Next, a sequencing experiment was performed only on MCF-7 cell line. Thereafter, the RNA from MCF-7, MDA-MB-231, and HaCaT were extracted and used to validate the sequencing data via qRT-PCR.

### 2.2. MicroRNAs Analysis of MCF-7 Cell Line in Response to trans-Chalcone

Following the sequencing assay, we conducted an analysis of the data, which took into account the quality of the sequencing, normalization data, and the microRNAs that were differentially expressed in MCF-7 cells treated with and without Tchal. The analysis identified 23 microRNAs that were modulated in MCF-7 cells following Tchal treatment. Of these, seventeen miRNAs were up-regulated and six miRNAs were down-regulated (Figure 1).

### 2.3. Validation by qRT-PCR

Five differently expressed miRNAs in MCF-7 cell line were selected for validation via qRT-PCR, two miRNAs oncogenic (miR-27a-3p and miR-891a-5p) and three miRNAs tumor suppressors (miR-25-5p, miR-449a and miR-4485-3p). In order to analyze whether some modulation could present a different pattern between cancer cell and non-cancer cell lines, we decided to perform validation of MicroRNAs selected using a normal keratinocytes cell line to compare the expression profile (Figure 2). Subsequently, when compared to HaCaT cells, all miRNAs exhibited statistically significant differences. Notably, miR-449a displayed sustained repression even after treatment with TChal, indicating its potential as a target for novel drug therapies. These findings suggest that targeting miR-449a could hold promise for developing effective treatments. Gene modulation values obtained by qRT-PCR demonstrate the reproducibility and accuracy of the sequencing.

### 2.4. Functional Categorization of Differentially Expressed microRNAs

Among the five miRNAs identified and validated in this study, all of them were found to be associated with genes that have been previously described in breast tumors. In this context, we identified fifty-six potential genes that could be targets of those miRNA. Then, a target miRNA–mRNA interaction network was assessed on Cytoscape, from which we can observe the interaction networks between the miRNAs and their predicted target genes (Figure 3).

### 2.5. Enrichment Analysis Results

The enriched signaling pathways are highlighted in Figure 4 and in the Appendix A. Among the top ten significant signaling pathways are pathways involved in ‘MicroRNAs in cancer’ with twelve of the fifty-six genes involved, ‘MAPK signaling pathway’ with nine genes, ‘Ras signaling pathway’, ‘PI3K-Akt signaling pathway’ both with eight genes, ‘Breast cancer’ with seven genes, and ‘Cellular senescence’ with six genes (Figure 4).

### 2.6. GO Enrichment Analysis Results

GO enrichment analysis was further employed to explore the enriched biological process (BP), molecular function (MF) and cellular component (CC). The top 10 that were significantly enriched according to an adj *p* < 0.05 identified are shown in Figure 5. Of the main BPs identified, ‘epithelial cell migration’, ‘epithelium migration’, and ‘tissue migration’ were among the more important ones according to the number of genes involved in these processes. For MFs, ‘protein tyrosine kinase activity’, ‘transmembrane receptor protein tyrosine kinase activity’ and ‘transmembrane receptor protein kinase activity’ were the main ones. For CCs, we highlight ‘cell leading edge’, ‘synaptic membrane’, ‘presynaptic membrane’ and ‘cell-cell junction’ (Figure 5).

### 2.7. In Silico Analysis of Gene Expression and Overall Survival

The expression levels of the *CSF1* and *KIT* genes were compared between breast cancer and adjacent non-tumor tissues from patients, using original published data available in the GEPIA database. The results showed that the expression of both *CSF1* and *KIT* was significantly downregulated in breast cancer tissues compared to non-tumor samples (Figure 6A,C). The association between the *CSF1* and *KIT* genes with patient OS was analyzed using the Kaplan–Meier plotter database. Based on the median expression of *CSF1* and *KIT*, patients were divided into high- and low-expression groups, and the OS of patients with low expression of both genes was significantly longer than that of patients with high expression of *CSF1* and *KIT* (Figure 6B,D).

## 3. Discussion

Our research group has been studying the antitumor effects of TChal in order to understand its mechanism and reveal its molecular targets. In some of these studies, TChal was able to act on the cell cycle, apoptosis, and metastasis [9,10,11]. Molecularly, it has been observed that microRNAs are important to many cellular processes and signaling pathways, including cell development, differentiation, proliferation, and apoptosis [16,17]. Also, the dysregulation of a single miRNA is able to bring many consequences to cells in terms of cellular outcomes and it can be associated with disease development, such as cardiovascular diseases, neurodevelopmental diseases, autoimmune disorders, bone diseases, and human cancers like breast cancer [16].

MiRNA sequencing of the MCF-7 cell line following TChal treatment revealed 23 miRNAs that were differentially expressed. Enrichment analysis revealed that all of the identified pathways were consistent with those previously identified by our research group, validating their relevance in cancer development. Additionally, the analysis also uncovered novel pathways that play a crucial role in cancer, including the MicroRNAs in cancer, breast cancer, MAPK, RAS signaling, TNF signaling, and PI3K-Akt pathways as shown in Figure 4. Based on their function in tumorigenesis and expression level, we selected five miRNAs (including oncogenic and tumor suppressor miRNAs) for validation and proceeded with the in silico analysis.

Two isoforms (miR-27a-3p and 5p) of miR-27a were identified. MiR-27a-3p isoform was selected as it is more frequently modulated in tumor cells [18]. The function of MiR-27a-3p in tumor development has been controversial because it can be both oncogenic and a tumor suppressor. However, in general, miR-27a-3p is associated with various aspects of tumor progression, including apoptosis regulation, cell migration, invasion, and proliferation [19]. During our in silico analysis, we identified several key genes associated with tumorigenesis such as *FBXW7*, *APPBP2*, *RUNX1*, *SPRY2*, *MAP2K4*, *NRP2*, *DNAJC13*, *CDR2*, *EFNB2*, *RAP1B*, *TEAD1*, *CCNK*, *RPS6KA5*, *MET*, *ATP2B1*, *RPS6KB1*, and *CSF1*. MiR-27a-3p has been implicated in tumorigenesis and suggests that the miR-23a/24-2/27a cluster may have a causal role in breast tumorigenesis by functioning as an oncogene that promotes cell invasion and metastasis. This occurs through the targeting of Sprouty2 (SPRY2) and subsequent activation of the p44/42 MAPK signaling pathway in breast cancer [20]. Sprouty2 (SPRY2) belongs to the highly conserved family of signal modulatory proteins that inhibit the Ras/MAPK pathway. It has been reported to be dysregulated in various cancer types, such as breast, liver, and prostate cancer, suggesting its potential role as a modulator of cancer progression. Specifically, SPRY2 has been implicated in regulating key pathways involved in cancer cell growth, migration, and invasion [20,21]. Another gene targeted by miRNA is colony-stimulating factor 1 (*CSF1*), which is responsible for attracting and differentiating macrophages into the M2 phenotype. In the development of breast cancer, the use of inhibitors targeting the colony-stimulating factor 1 receptor (CSF1-R) can eliminate TAMs, leading to effective inhibition of metastasis and angiogenesis, as well as reduction in tumor invasiveness [22]. Tumor-associated macrophages (TAMs) are known to secrete cellular cytokines and surface receptors that play a critical role in promoting breast cancer metastasis. One of the key factors is the high expression of epidermal growth factor (EGF) in TAMs, which activates epidermal growth factor receptors (EGFRs) in cancer cells, leading to metastasis and increased secretion of colony-stimulating factor 1 (CSF-1). CSF-1, in turn, recruits and activates TAMs to further secrete EGF, indicating the presence of an EGF/CSF-1 positive feedback loop between TAMs and cancer cells. The EGF signaling pathway also induces the infiltration of breast cancer cells into blood vessels, thereby promoting blood vessel metastasis. These findings highlight the importance of TAMs in breast cancer progression and provide potential therapeutic targets for inhibiting metastasis [23,24]. Interestingly, the *EGFR* inhibition by TChal on MCF-7 and BT-20 cell lines had previously been proved by our research group [10]. Another study by Li et al. demonstrated that overexpression of miR-27a-3p led to *EGFR* inhibition, which is related to tumor growth, and induced changes in the cellular process such as those stated above [25]. Then, we suggested that the up-regulation of miR-27a-3p could inhibit the *EGFR* gene and *CSF1*. Moreover, when the OS patients were analyzed, the one who presented the lowest expression of the *CSF1* gene was significantly longer than that of patients with high expression with a HR = 0.81.

Likewise, in vitro and in vivo assays have shown that modulation of miR-27a-3p can significantly inhibit the proliferation and invasion of ESCC cells in nude mice [26]. Furthermore, it is also involved in the regulation of the *ZBTB10* gene, which is related to up-regulating Sps proteins important to tumor survival and angiogenesis [27]. Moreover, we have demonstrated that Sp1 was downregulated by TChal [11]. Additionally, the Heme-Oxygenase 1 (HO-1) is up-regulated at a protein level by TChal, and this is able to reduce breast cancer in xenograft assays with nude mice [12].

MiR-449a is often repressed in breast cancer and is associated with improved cell proliferation, migration, invasion, and metastasis in cancers such as breast cancer, hepatocarcinoma, gastric cancer, prostate cancer, and endometrial cancer [28,29]. In addition, the suppression of this microRNA is negatively associated with aggression and progression in breast cancer [30]. It occurs due to the modulation of genes that lead to the activation of resistance (*ADAM22*), the cell cycle (*CDC25*), proliferation and metastasis (*PLAGL2* and *TPD52*) [28,29,31,32]. Hence, these data suggest that drugs that can increase the expression of miR-449a could lead to a decrease in tumor progression. Thus, TChal has been shown to be an alternative in the treatment of breast cancer, as it significantly induced the expression of this miRNA in MCF-7, whereas, in the HaCat, normal cell, this miRNA remained repressed. One of the gene targets of this microRNA is *MDM4*. Previously, our group observed that the *MDM4* gene was induced in the MCF-7 cell line after treatment with both *trans*-chalcone and licochalcone A; however, in the BT-20 cell line, this gene was induced only by Licochalcona A [10]. The *MDM2* and *MDM4* genes are p53 inhibitors and are frequently induced in cancers [33]. Although some tumors have high levels of *MDM4* modulation and mutations in the *TP53* gene, this mutation may confer a cooperativity of both genes in tumor development. Approximately 14% of invasive breast tumors have *MDM4* amplification and 3% have *TP53* gene mutation [34]. In addition, authors have suggested that *MDM4* is able to act independently of *p53*, due to its interaction with other proteins such as p21, 14-3-3gamma, ARF, HAUSP and Nbs1, these interactions could cause genetic instability and increase cell transformation [34,35,36,37]. Therefore, our research group has proposed that TChal exerts its effects by inducing the expression of *TP53*, a tumor suppressor gene, while suppressing p21, an oncogenic gene [11]. These findings suggest that the regulation of the *MDM4* gene, which is known to interact with *TP53*, may occur independently from *TP53* itself. Considering that the relationship of modulation of microRNAs to their target genes is usually inversely proportional, we suggest that miR-25-5p repression may positively modulate the *MDM4* gene.

During the in silico analysis, two miRNAs, miR-27a-3p and miR-449a, were found to modulate the *MET* gene. The MET receptor tyrosine kinase is known to activate various cellular functions that are crucial to organ development and cancer progression upon binding with its ligand, hepatocyte growth factor (HGF). Abnormal c-Met signaling has been observed in different types of cancer, making the receptor a promising therapeutic target [38]. Although a high number of *MET* copies did not independently predict recurrence-free survival, the researchers did observe lower recurrence-free survival rates in the group with amplified MET during univariate analysis. Furthermore, a positive correlation was found between the number of *MET* copies and triple-negative status [39].

The in silico analysis revealed that miR-25-5p targets *KIF22*, *GPX1*, *BCAR1*, *SKP2*, *SLC2A12*, *FSCN1*, *CYCS*, *NTRK2,* and *ETS1* genes. MiR-25-5p is known to promote cancer cell proliferation, anti-apoptosis, cell cycle progression, and cell motility, and is considered an oncogenic miRNA [40,41]. It has been shown that miR-25-5p is induced by the WNT/β-catenin pathway and promotes epithelial–mesenchymal transition (EMT) and metastasis by regulating the RhoGDI1 gene [42]. Moreover, the interaction between mutant *TP53* and *BCAR1* activates downstream signaling pathways that contribute to cancer cell invasion [43]. Therefore, inhibiting miR-25-5p with drugs like TChal could be a potential therapeutic strategy for breast cancer.

miR-4485 is related to mitochondrial functions, although it is synthesized in the cell nucleus, it is transported to the mitochondria. This miRNA is involved in ATP production processes, ROS level, cell cycle, activation of caspases 3/7, and apoptosis [44,45]. In addition, Sripada et al. 2017 proved the influence of miR-4485 on tumor development by performing miR-4485 mimic transfection, leading to glycolytic pathway regulation and reduced chlorogenic ability of MDA-MB-231 cells, and they also observed a reduction of tumorigenicity in vivo [45]. Otherwise, the in silico data demonstrated that the gene *KIT* is a target by the identified miRNA. The *c-Kit* gene has been shown to be important in breast cancer, particularly in triple-negative breast cancer (TNBC), which lacks targeted therapies. Studies have shown that *c-Kit* is involved in promoting the migration and invasion of TNBC cells, and that inhibiting *c-Kit* with tyrosine kinase inhibitors may be a promising treatment option. Further research is needed to fully understand the role of *c-Kit* in breast cancer and to develop effective targeted therapies [46,47,48]. Moreover, when the OS patients was analyzed the one whose presented the lowest expression of *KIT* gene was significantly longer than that of patients with high expression with a HR = 0.69. These data indicate that miR-4485 could negatively affect tumor development, suggesting that this gene is a tumor suppressor in breast cancer.

Finally, miR-891a-5p has been identified as a potential biomarker for breast cancer as it is highly modulated by TChal. This microRNA is located on chromosome Xq27.3 along with 7 other microRNAs (miR-892, miR-890, miR-888, miR-892b, miR-891b and miR-891a) and has been identified as proto-oncogenic. The target gene of miR-891a-5p is TIMP2, which is associated with proliferation, migration, and tumor formation in prostate cancer [49]. In addition, our in silico analysis showed that it can modulate *CNR1* and *PTCH1* genes. Zhang et al. showed that miR-891a-5p inhibited the expression of *ADAM10* by targeting its 3’UTR. This inhibition resulted in the suppression of proliferation and migration of breast cancer cells, particularly those that are Hormone receptor-positive [50]. The miR-891a-5p overexpression can negatively regulate the *HOXA-5* gene (Homeobox A5), known as a tumor suppressor, related to the regulation of *TP53* [51].

## 4. Materials and Methods

### 4.1. Materials

The *trans*-Chalcone (TChal) and some of the main reagents including trypsin, doxorubicin hydrochloride, dimethyl sulfoxide (DMSO), DMEM, penicillin, kanamycin, streptomycin, and Fetal bovine serum were purchased from Sigma-Aldrich (St. Louis, MO, USA). All the materials used for sequencing as Taqman microRNA Reverse Transcription kit and Taqman Small RNA Assays were both purchased from Applied Biosystems and MiSeq v2 Reagent kit (Illumina, San Diego, CA, USA). Finally, the materials to gene expression assay TaqMan^TM^ MicroRNA Assay and TaqMan^TM^ Universal Master Mix II, no UNG were from Thermo Fisher, Waltham, MA, USA.

### 4.2. Cell Culture

The breast cancer cell lines MCF-7 (Luminal A, estrogen receptor (ER) positive), MDA-MB-231 (triple-negative breast cancer), and HaCat (human keratinocyte cell line) were grown in DMEM medium supplemented with 10% fetal bovine serum, 100 U/mL penicillin, 100 μg/mL streptomycin, and 100 μg/mL kanamycin. The cultures were maintained in a humidified atmosphere with 5% CO${}_{2}$ at 37 °C until 90% confluence.

### 4.3. Viability Assay Conditions

The inhibition of viability cells induced by TChal was evaluated by MTT assay on MCF-7, MDA-MB-231, and HaCat (non-cancerous cell line). This test was performed as described by Komoto et al. [8] with modifications. Briefly, each cell line was seeded at a density of 2 × 10^3^ cells/well in 96-well plates and incubated for 24 h. The cells were treated with 7 different concentrations of TChal (100, 50, 25, 12.5, 6.25 μM) dissolved in 0.1% DMSO for 24 h. Next, 20 μL MTT solution (5 mg/mL) was added to each well, and the plates were incubated for an additional 3 h. The formazan dye was diluted in 200 μL DMSO. Absorbance was verified at 550 nm in a microplate reader MultiSkan FC. The assays were carried out in three independent experiments performed in triplicate. Doxorubicin was used at 2.5 μg/mL (4.31 μM) as the positive control for this test. The percentage of inhibition of cell viability and the IC50 calculation was based on Komoto et al. [10].

### 4.4. cDNA Library Construction and Mi-seq Assay

The total RNA from both MCF-7 cells treated with TChal and MCF-7 control treated with DMSO were extracted using the miRNeasy Isolation (Qiagen, Germantown, MD, USA) and the preparation of the miRNA library used the TruSeq Small RNA Library (Illumina, San Diego, CA, USA) kit, and both assays were done following the manufacturer’s instructions. The integrity of each miRNA library was analyzed by a bioanalyzer (Agilent Technologies, Santa Clara, CA, USA) as per the manufacturer’s instructions. For the library construction, the total RNA sample was used to the manufacturer’s instructions, and adaptors were added to the small RNAs. The small RNAs were converted into cDNAs by PCR, then to obtain the mature miRNA, electrophoresis was performed using the polyacrylamide gel (12%), and the mature miRNAs were between 145–160 pb, so the fragments were cut off the gel. Afterward, the fragments were purified, and the cDNA library was quantified by KAPA library quantification (Illumina sequencing platforms). Lastly, the samples were sequenced using a MiSeq v2 Reagent kit (Illumina, San Diego, CA, USA) on a Mi-seq system.

### 4.5. Validation Data

The validation data were performed by qRT-PCR using primes TaqMan™ MicroRNA Assay (Thermo Fisher, Waltham, MA, USA) following the manufacturer’s instruction, the selected miRNAs were: miR-25-5p, miR-27a-3p, miR-891a-5p, miR-449a, and miR-4485-3p, with RNU44 used as a housekeeping microRNA gene. RNU44 was chosen because it is present in more than 219 breast cancer types [52]. The entire experiment was carried out in three independent experiments using the TaqMan™ Universal Master Mix II, no UNG kit (Thermo Fisher, Waltham, MA, USA) and the machine Mx3300 qPCR System (Stratagene, San Diego, CA, USA) for MCF-7 (breast cancer) and normal cell line HaCaT (keratinocyte non-cancerous cell) after treatment with TChal at IC50 pre-established concentrations.

### 4.6. Bioinformatic Analysis

Raw miRNA sequences were initially submitted to a quality filter (*q* < 28) using Trimmomatic v.0.36 and small RNA sequencing adapters were removed using CutAdapt v.1.4.1. Alignments were performed using Bowtie2 v.2.1.0 and the counts of reads mapped to the human miRNA dataset in miRBase, Release 20 were obtained according to HT-Seq v. 0.7.2. Differential expression analysis (DEA) was performed using the DeSeq2 package. Differentially expressed miRNAs were considered with a false discovery rate (FDR) cut-off of 0.1 (10% FDR) and fold-change ≥ 2.0. The comparisons of the miRNA profiles were visualized according to heatmaps and constructed using the Complex Heatmaps package.

### 4.7. Prediction of microRNA Targets

To evaluate the biological role of miRNAs, bioinformatics analysis was initially applied to predict their targets using the free and online program miRDIP (http://ophid.utoronto.ca/mirDIP/, accessed on 13 April 2023), with a focus on the Top 1% predicted target genes (score class very high). This program integrates dozens of miRNA target prediction tools, each using a specific algorithm, thereby increasing prediction accuracy. From the generated lists of genes, targets predicted by at least three of the following prediction algorithms were selected: DIANA, microrna.org, RNA22, RNAHybrid, and TargetScan.

### 4.8. Enrichment Analysis

To assess the potential cooperative effects of identified miRNAs on biological processes and pathways associated with breast cancer, we performed pathway enrichment analyses using Reactome and its Cytoscape plugin that provides curated regulatory interaction networks and biological pathways derived from Reactome and other databases. We selected genes associated with breast cancer, according to the Cancer Gene Index Annotations (provided by the National Cancer Institute, NCI), by applying the Load Cancer Index function available in Reactome. We only considered pathways with FDR values ≤0.001 or less and at least three genes involved to increase the accuracy of the analysis. Enrichment analysis maps were obtained from an online data analysis website (http://www.bioinformatics.com.cn/, accessed on 13 April 2023).

### 4.9. Gene Expression Profiling and Interactive Analysis (GEPIA2)

Gene Expression Profiling and Interactive Analysis (GEPIA2) is an online tool for gene expression analysis (http://gepia2.cancer-pku.cn/index, accessed on 13 April 2023). This tool allows performing differential analyses with the gene expression profile of the tumor in relation to normal TCGA or GTEx data. We used this server to analyze target gene expression for the miRNAs identified in our study. GEPIA2 performs differential expression analysis and returns the result in a boxplot [53].

### 4.10. Overall Survival (OS) Analysis

An overall survival (OS) analysis was performed using the Kaplan–Meier plotter database (http://kmplot.com/analysis/, accessed on 13 April 2023) to evaluate the OS of target genes in breast cancer. Kaplan–Meier Plotter is an online database containing survival and gene expression information for various types of cancer. We selected only data available for breast cancer. The analysis compares high and low gene expression, and the results were shown by log-rank and *p*-value.

### 4.11. Statistical Analysis

All experimental data were submitted to analysis of variance (ANOVA) followed by Bonferroni test. The level of statistical significance was set at *p* ≤ 0.05.

## 5. Conclusions

In conclusion, this research has provided valuable insights into the therapeutic potential of TChal for breast cancer by shedding light on its potential molecular targets. The miRNA sequencing and enrichment analysis employed in this study have identified several crucial pathways involved in tumorigenesis, including miRNA in cancer, breast cancer, MAPK, RAS signaling, TNF signaling, and PI3K-Akt pathways. These findings underscore the importance of TChal in breast cancer treatment. Furthermore, the in silico analysis conducted in this research has revealed the targeting of several significant genes associated with tumorigenesis, such as *CSF-1* and *KIT*. Importantly, it was observed that lower expression of these genes correlates with better overall survival. These findings highlight the potential of TChal to inhibit tumor progression and metastasis in breast cancer. Overall, this research contributes to our understanding of TChal’s therapeutic potential and provides a foundation for further investigations and the development of targeted therapies for breast cancer. By elucidating the molecular mechanisms and pathways affected by TChal, this study paves the way for future studies and clinical trials aimed at harnessing the benefits of TChal in breast cancer treatment.

## Figures and Tables

**Figure 1 ijms-24-10785-f001:**
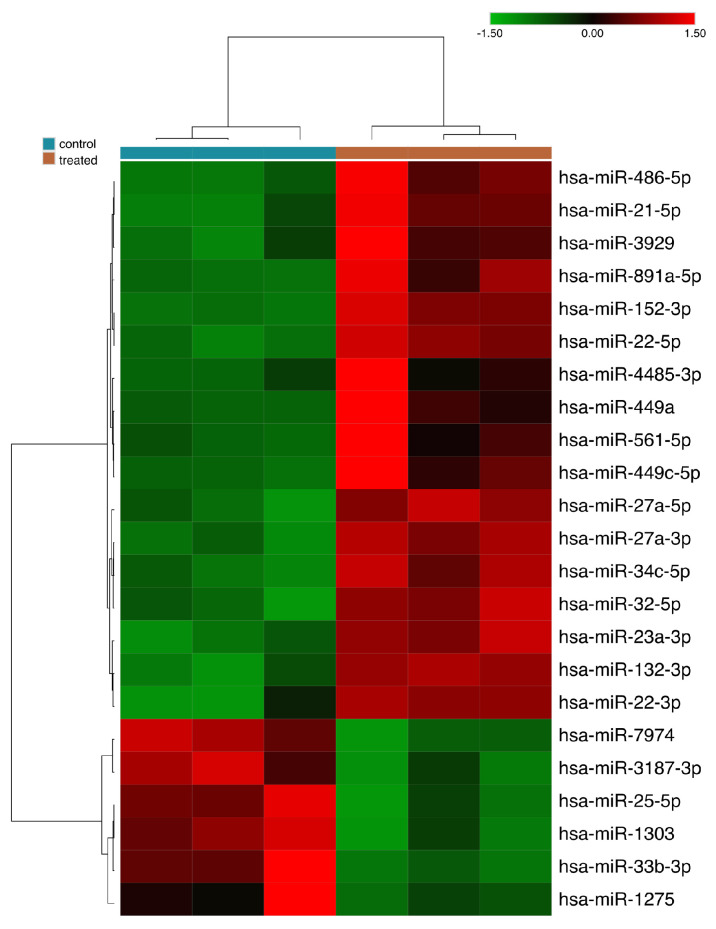
Differentially expressed microRNAs of MCF-7 treated and non-treated with Tchal. The IC50 was used for this experiment.

**Figure 2 ijms-24-10785-f002:**
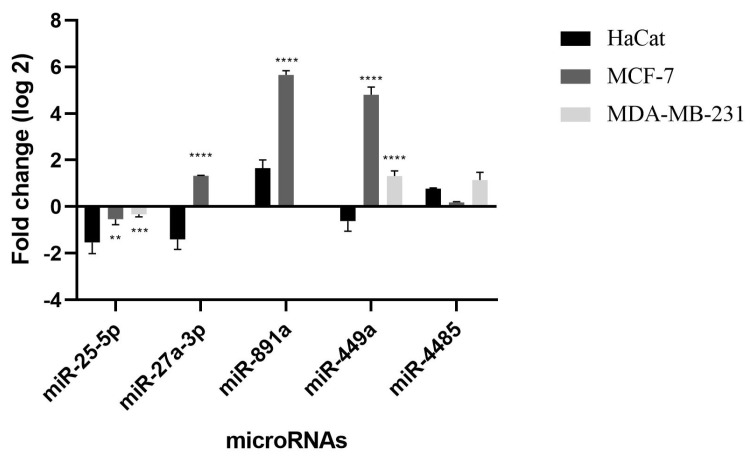
Validation of five miRNAs differentially expressed after treatment with TChal at IC50 in MCF-7, MDA-MB-231 and HaCat cell lines. Considering ** *p* < 0.01, *** *p* < 0.001, **** *p* < 0.0001.

**Figure 3 ijms-24-10785-f003:**
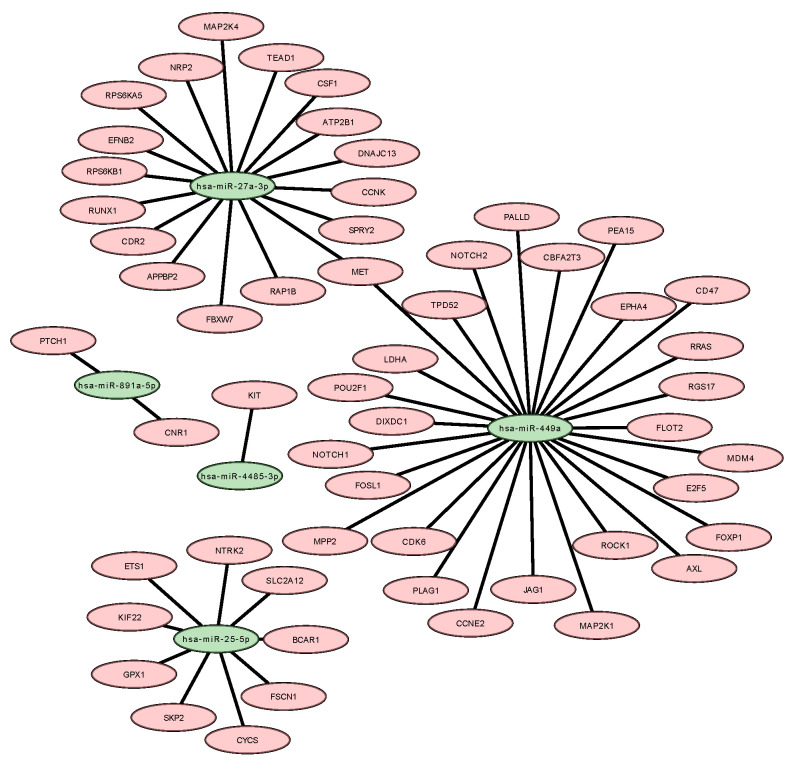
Based on in silico, an interaction network between miRNA and their respective gene target was designed, as it demonstrated the gene MET could be regulated by two microRNAs.

**Figure 4 ijms-24-10785-f004:**
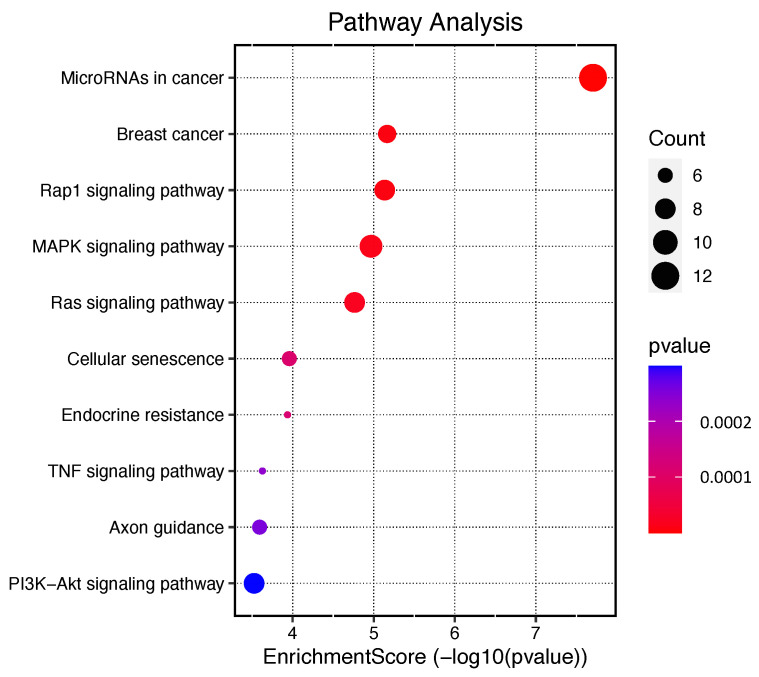
Dotplot-enriched signaling pathways were used to demonstrate the primary biological actions of key potential targets. Bubble size indicates the number of deferentially expressed genes in the corresponding pathway. The color indicates value −log10(lowest *p*); the more it shifts to red, the more significant the pathway is.

**Figure 5 ijms-24-10785-f005:**
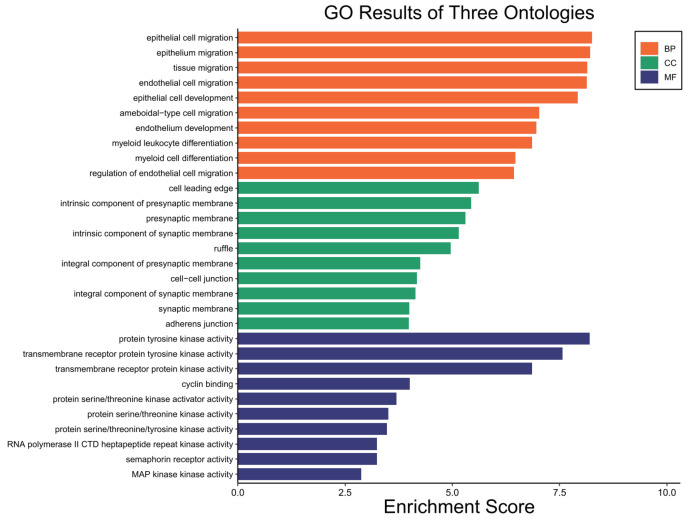
Enrichment analysis considering biological processes (BP), cellular components (CC), and molecular functions (MF) of target genes from the pre-selected miRNA.

**Figure 6 ijms-24-10785-f006:**
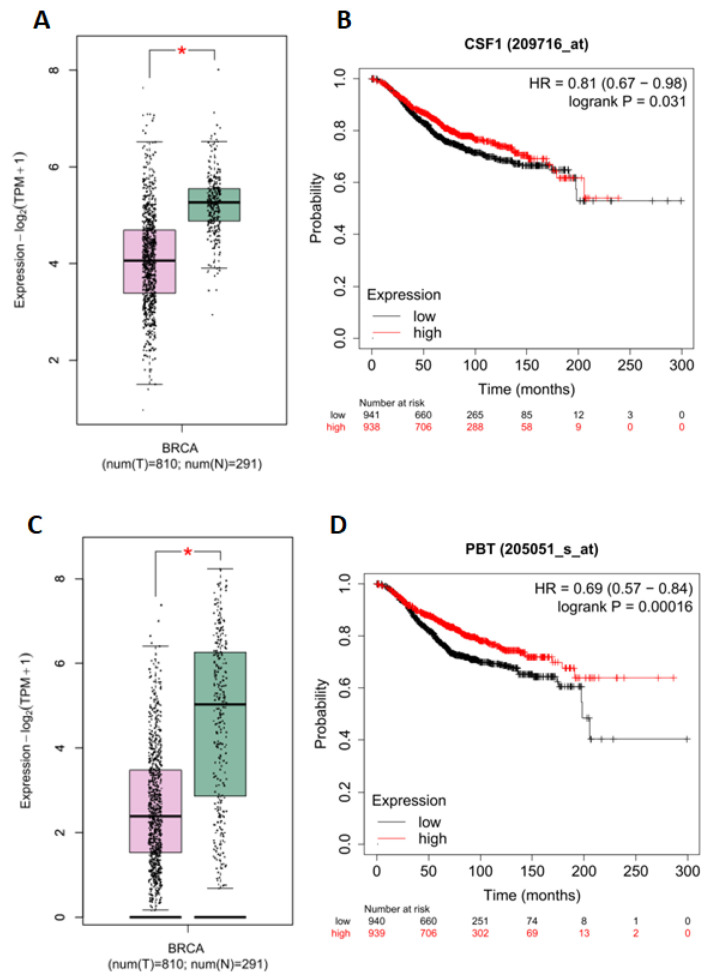
In silico analysis. (**A**) Analysis based on the GEPIA 2 database showed low expression of *CSF1* in breast cancer tissues (BRCA). (**B**) The association between *CSF1* expression and overall survival of breast cancer patients was analyzed using the Kaplan–Meier plotter database. A log-rank test was performed to compare the groups. *n* = 1879; *p* = 0.031. (**C**) The GEPIA 2 database analysis revealed high expression of *KIT* in breast cancer tissues (BRCA). (**D**) The Kaplan–Meier plotter database was used to analyze the association between *KIT* expression and the overall survival of patients with breast cancer. A log-rank test was performed to compare the groups. *n* = 1879; *p* < 0.00016. For both genes, *CSF1* and *KIT*, the boxplot analysis revealed statistical significance with * *p* < 0.05.

**Table 1 ijms-24-10785-t001:** The IC50 (μM) of *trans*-chalcone against breast cancer and normal cell lines.

Cell Lines	Concentration (μM)
MCF-7	53.73
MDA-MB-231	29.06
HaCat	44.18

## Data Availability

The data presented in this study are available on request from the corresponding author.

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
