# Peer review of "Exploring the Therapeutic Potential of trans-Chalcone: Modulation of MicroRNAs Linked to Breast Cancer Progression in MCF-7 Cells"

_ijms, 2023, doi:10.3390/ijms241310785_

Round 1
Reviewer 1 Report
Dear Authors,
From the manuscript, a nice effect on the expression of miRNAs seems to happen after the treatment of MCF-7 cells with TChal. It is not fully clear with me why/how 5 of those miRNAs were chosen for further analysis. At least, in the discussion part there is a detailed review of their potential way of action in breast cancer. I recommend some changes for text and scientific clarity.
My main concerts are listed in the major comments part.
Title: In first, I was wondering if the title is proper. The title nicely states the findings in figure 1.
I would kingly suggest that certain parts of the text could require rephrasing so clarity will be improved:
· Lines 4-5
· Sentence at lines 10-11, missing scientific soundness
· Sentence at lines 21-23
· Sentence at lines 30-34
· Line 36-37: ‘the main miRNA function is a post-translational gene regulation of….’, sounds general. A possibly alterative way to state that could be: ‘The main function of miRNAs is the post- translational gene regulation of….’
· Lines 76-77: Maybe rephrase the respective part to ‘ treated with TChal and non-treated (Control-DMSO only’
· Lines 94-95: ‘as well as having been associates with a poor prognosis’, changes related to the used tense
· Line 132: statement ‘performing differential analyses’, -ing seems not to be the proper way
· Legend of figure 1: could be nice to mention either the concentration of TChal in the case of MCF-7 or something like that ‘ with the respective for MCF-7 concentration of Tchal’
· Sentence at lines 169-171
· Sentence at lines 174-175
· Legend of figure 5: add also the abbreviations BP, CC, MF at the legend
· Sentence at lines 217-220
· Line 226: part ‘In general, it is..’
· Line 230: part ‘ and a suggest that ..’
· Sentence at lines 227-230: seems that a verb is missing.
· Line 2457: is receptor (EGFR)
· Line 255: Rephase the part ‘27 demonstrated’
· Line 267: ‘the tumor’, what type of tymor?
· Sentence at lines 289-291
· Line 303: could you state the meaning of ‘TN status’?
· Sentence at lines 313-316
· Sentence at lines 319-320
· Sentence at lines 326-328
· Line 336: ‘A Zhang, 2020’. Rephrase that part
I would strongly suggest to double-check for misprints, for example:
· Space missing: Lines 9, 28, 154, 224
· Line 10: comma before respectively
· Double space: lines 103, 131,
· Extra fullstop: line 204
· Line 308: it is β-catenin
· Line 42: ’and others’. What that statement means?
· Line 56: the description of MCF-7 in the parenthesis
· Line 57: maybe cells instead of ‘non-cancerous cell’
· Line 112: alignment of the line
· Line 150: ‘from MCF-7’, instead of ‘to MCF-7’
· Table 1: choose between μΜ and uM
· Capital F: figure 2, line 168; figure 3, line 178
· Replace figure x, with number: lines 180, 188
· Line 176: ‘miRNA-RNAm’, could possibly mean ‘miRNA-mRNA’?
· Line 310: contributes
· Line 318
Other scientific and text related questions for clarity improvement:
· Line 31: ‘against cancer’. What type of cancer?
· Line 34: ‘mediated by Heme-Oxigenase 1’. What is the mechanism? It is referred in the text, but in a letter point
· Line 35: ‘mirna is an extensive class’, the same is repeated in the abstract
· Lines 40-41: instead of ‘as a biomarker’, an alternative could be ‘as biomarkers’
· Figure 4: Clarity what key potential targets. Whose key potential targets?
· Line 208: How TChal could affect the indicated functions? Positively, negatively?
· Analysis of GEPIA 2 in figure 6: that type of analysis correlates in the pool of BRCA patients, meaning independently of ER status?
Major scientific comments
· Paragraph 2.3: MDA-MB-231 cells are not included.
· Line 74: SD is not provided for the IC50 in the text.
· Paragraph 2.4: I am not familiar with the indicated workflow to identify miRNAs. I would like to know why the approach for sequencing of miRNAs was used. The sequence of the miRNAs is not known? I am only familiar with other approaches that amplify miRNAs, create cDNA for a specific miRNA and detect it with PCR.
· Could you provide a better analysis of images used in figures 3, 4 and 5?
· May I kindly request further clarification regarding the data presented in the manuscript? Specifically, the statement that TChal was a similar IC50 for MCF-7 and MDA-MB-231, with both values being a little bit higher than the IC50 of non-tumor cells. It appears that TChal may be more toxic for healthy cells rather than breast cancer cells, but without a corresponding viability curve graph to examine, I am unable to verify this.
· Also, the comparison at figure 2 does not help to understand the miRNA status before and after the treatment with TChal in each cell line. Perhaps, a more clear description of the reason this comparison is important will help the reader.
· It is not clear with me why the genes CSF1 and KIT were chosen and how they are important for the analysis. Maybe that point needs clarification.
· May I kindly request further clarification on the experimental design presented in the manuscript? I am curious why the differential analysis in fig 1 was only conducted in MCF-7 cells and not in MDA-MB-231 and non-tumor cells, as well. One step ahead, I am missing the point of how the 5 DE miRNAs are chosen from the MCF-7 analysis and validated in the other cell types. Not all miRNAs are important for non-tumor vs ER-positive vs triple negative cells, so TChal could act in a different way in the 3 used cell lines.
· I was wondering if you could provide some additional clarification on the conclusion drawn in the manuscript. Based on the findings presented, I am uncertain if the conclusion is fully supported. Specifically, I am unsure if the manuscript effectively sheds light on the potential molecular targets of TChal for breast cancer treatment. I would appreciate a realistic and evidence-based assessment of the manuscript's contributions to the field
· In figure 1, there is a clear swift between all the studied miRNAs in MCF-7. This figure seems to be important due to that clear effect. If you would like, you could add a few words about that effect and the importance of that observations.
· Are there any future perspectives of this study that seem to be important to be mentioned in the text?
· Usually, it is descripted that such compounds have a biological activity when the IC50 is less than 10μΜ. In the case of TChal the IC50 value is much higher but it seems that there is some significant effect in the expression of miRNAs. Do you believe that this is a point that you would like to address in the manuscript?
Good Luck
In total the quality of the English Language is adequate. I recommend some changes in the main comment sections for clarity.
Reviewer 2 Report
The authors have demonstrated an interesting research finding on the effects of Trans-Chalcone on miRNAs expression and its relationship with breast anticancer activity. The work is well planned and executed.
However there are some minor comments which the authors need to address/rectify.
1. In line 46-47 doxorubicin hydrochloride would be a single word, please correct it.
2. The figure 3 is not readable, please increase the font size of the texts.
3. Could these 5 mRNA (miR-25-5p,12 miR-27a-3p, 12 miR-891a, miR-449a, and miR-4485) selected for this study have the potential to act as molecular targets for most of the breast cancer treatment drugs or are these specific to Trans-Chalcone?
4. The text in the x-axis of figure 6 B & D is also not readable and the size needs to be increased.
5. As miRNA could be a potential target for therapeutic similarly miRNA have been closely associated to cancer drug responses and patient outcome after chemotherapy. A similar study with miRNA 195 has demonstrated the fact (https://doi.org/10.3390/cancers13235979). The author should include this in their discussion as miRNA could also act as a predictive markers for cancer drug responses.
Round 2
Reviewer 1 Report
Dear Authors,
thank you for your time answering the review report. Your previous published article is of great support for this manuscript as of the importance of TChal.
The conclusion section is nicely enriched.
Please, be sure that all figures are in place, for example, figure 3 in the revised edition.
Wish you the best.